# Serum IgE and IgA Levels in Pediatric Henoch–Schönlein Purpura: Clinical Characteristics and Immunological Correlations in the Context of Infectious Diseases—A Five-Year Retrospective Analysis

**DOI:** 10.3390/ijms26136053

**Published:** 2025-06-24

**Authors:** Sînziana Oprițescu, Gabriela Viorela Nițescu, Mihaela Golumbeanu, Dora Boghițoiu, Elena Iuliana Ioniță, Monica Licu, Larisa-Marina-Elisabeth Chirigiu, Violeta Popovici, Loredana-Maria Marin, Elena Moroșan

**Affiliations:** 1Discipline of Clinical Laboratory and Food Safety, Faculty of Pharmacy, “Carol Davila” University of Medicine and Pharmacy, 6 Traian Vuia Street, 020945 Bucharest, Romania; sinziana.opritescu@drd.umfcd.ro (S.O.); elena.morosan@umfcd.ro (E.M.); 2“Grigore Alexandrescu” Clinical Emergency Hospital for Children, 017443 Bucharest, Romania; 3Discipline of Pediatrics, Faculty of Dentistry, “Carol Davila” University of Medicine and Pharmacy, 020021 Bucharest, Romania; 4Discipline of Pharmacognosy, Phytochemistry and Phytotherapy, Faculty of Pharmacy, “Carol Davila” University of Medicine and Pharmacy, 6 Traian Vuia Street, 020945 Bucharest, Romania; 5Department of Medical Psychology, Faculty of Medicine, “Carol Davila” University of Medicine and Pharmacy, 050474 Bucharest, Romania; 6Faculty of Pharmacy, University of Medicine and Pharmacy Craiova, Petru Rares, 2, 200349 Craiova, Romania; 7Center for Mountain Economics, “Costin C. Kiriţescu” National Institute of Economic Research (INCE-CEMONT), Romanian Academy, 725700 Vatra-Dornei, Romania; 8Department of Pharmaceutical Physics, Faculty of Pharmacy, “Carol Davila” University of Medicine and Pharmacy, 020956 Bucharest, Romania

**Keywords:** Henoch–Schönlein purpura, IgA vasculitis, allergic vasculitis, infectious diseases, hyper IgE

## Abstract

Immunoglobulin A vasculitis (IgAV), previously known as Henoch–Schönlein purpura (HSP), is a type of non-thrombocytopenic small-vessel vasculitis. HSP is the most common systemic vasculitis in pediatric patients, and it is characterized by purpura, arthritis or arthralgia, gastrointestinal pain, and renal dysfunction. This retrospective analysis also examines a range of demographic factors, including sex, geographic and environmental influences, age, and medication, to evaluate their potential effects on the pediatric population affected by HSP. The five-year hospital-based retrospective analysis included 138 hospitalized children diagnosed with HSP during hospitalization. Blood sample analysis was conducted to assess various immunological parameters, including levels of immunoglobulins (IgA and IgE), complement components (C3 and C4), C-reactive protein, fibrinogen, the erythrocyte sedimentation rate (ESR), and allergen panels. Elevated IgE levels and normal IgA serum concentrations were found to be strongly associated with infectious diseases in pediatric HSP patients. Patients with recurrent infectious diseases consistently exhibited elevated IgE levels and normal IgA levels during treatment despite no identified allergens, alongside an increased risk of disease recurrence.

## 1. Introduction

Immunoglobulin A (IgA) vasculitis (IgAV), also known as Henoch–Schönlein purpura (HSP), is the most common systemic vasculitis in children, characterized by purpura, arthritis or arthralgia, gastrointestinal symptoms, and renal involvement [1,2,3,4,5]. It typically affects children between 3 and 15 years of age and occurs in 3–27 cases per 100,000 children, being rare in infants (age 0 to 2 years) [1,5,6,7].

Although the exact mechanisms underlying HSP are not fully understood, the disease is linked to immune dysregulation, notably the deposition of IgA-dominant immune complexes in small blood vessels [8]. Observations such as the accumulation of IgA-dominant immune complexes in blood vessels, the presence of specific white blood cells in small blood vessels, and the degradation of these white blood cells indicate the immunological nature of this disorder. IgA, a key immunoglobulin in mucosal immunity, is produced in large quantities and has a short half-life [9,10]. Various triggers, such as infections, insect bites, vaccines, medications, and food allergies, have been associated with the onset of HSP [9,11,12,13,14,15]. While IgA plays a central role in disease pathogenesis, other immune factors may also contribute.

Emerging evidence points to a possible role of immunoglobulin E (IgE) in HSP, especially in pediatric patients with recurrent infections, but no clear allergic sensitization [14]. Investigating the involvement of IgE could provide valuable insights into the triggers and underlying immune pathways of HSP, potentially improving our understanding of disease recurrence and progression.

Immunoglobulin E (IgE) plays a well-established role in defending the body against parasites and environmental toxins, also is primarily known for mediating allergic reactions such as rhinitis, asthma, eczema, and anaphylaxis [16,17,18,19]. Elevated IgE levels are commonly seen in allergic diseases and used as markers of atopy in children [20]. However, high IgE may also be present in other conditions, including infections, autoimmune diseases, and rare immunodeficiencies like hyper-IgE syndrome [21]. Notably, some children with recurrent infections show elevated IgE without identifiable allergic triggers, and IgE levels tend to peak in childhood and adolescence before declining with age [16,22,23].

In the context of HSP, the association between recurrent infections and elevated IgE suggests that immune-allergic mechanisms may contribute to disease development and recurrence [14,24,25,26]. Furthermore, exposure to environmental irritants can raise IgE levels, leading to respiratory symptoms [27]. Although HSP is primarily characterized by IgA-mediated vasculitis, research findings suggest that IgE may also contribute to the immune response related to the onset and recurrence of the disease [14,16,22,26,28,29]. This underscores the need to further investigate the immunological profile of HSP patients, particularly the interplay between IgE, IgA, and infectious triggers, to gain a clearer understanding of the disease’s underlying mechanisms.

This study aims to gain a deeper understanding of the clinical characteristics of children diagnosed with HSP by investigating the potential role of infectious diseases and specific immune markers—namely, IgE and IgA—in the development of this condition. In addition to examining immune factors, this retrospective study also investigates whether factors such as age, sex, living environment, and medication use may influence the presentation and progression of HSP in young patients. One of the primary objectives is to elucidate the potential role of IgE in this context, particularly since its association with HSP has been less well-established compared to IgA.

## 2. Results

### 2.1. Descriptive Statistics

A retrospective study conducted using a five-year hospital record included 138 pediatric patients diagnosed with HSP, with 81 (58.69%) being male and 57 (41.30%) being female. The mean age of the cohort was 7.39 ± 3.86 years old, with females being slightly older than males (6.96 ± 0.411 years old vs. 8.00 ± 0.536 years old). The average age of the group was 7.39 ± 3.86 years. The results from the Kolmogorov–Smirnov and Shapiro–Wilk statistical tests indicate that the age distribution for both genders does not follow a normal distribution (*p* < 0.01 for both sexes). The median age of the entire cohort was 7 years old, with females averaging 7 years old and males averaging 6 years old. The interquartile range (IQR) for the overall group was 6 years old, with an IQR of 6 years old for females and 5 years old for males. The ratio of urban to rural patients was 1.46:1, with 82 urban patients and 56 rural patients.

Of the 138 patients included in the study, 65.21% presented with moderate disease, predominantly among those from urban areas. Mild symptoms were observed in only 7.97% of cases, representing the smallest subgroup. Severe forms of HSP were identified in 26.81% of patients, with an almost equal distribution between urban and rural environments. Among those with severe disease, 63.8% were male. Figure 1 presents the distribution of HSP severity levels concerning the patients’ environmental background.

The study analyzed hospitalization dates to explore potential associations between HSP onset and seasonal factors such as allergens or infections. Autumn accounted for the highest number of cases, with 40 patients (28.98%). Winter and spring showed comparable hospitalization rates, with 35 and 37 cases, respectively. Summer had the lowest number of hospitalizations, with 26 patients (18.84%), and notably, no mild cases were recorded during this season. Figure 2 illustrates the seasonal distribution of HSP cases concerning disease severity.

### 2.2. Immunologic Panel Results

#### 2.2.1. Principal Component Analysis Results

We examined potential correlations between serum IgA and IgE levels in the patient cohort. We included inflammatory markers—complement C3, complement C4, fibrinogen, erythrocyte sedimentation rate (ESR), and C-reactive protein (CRP)—in a principal component analysis (PCA). The aim was to investigate the relationship between immunoglobulin levels and systemic inflammation. However, the initial principal component analysis (PCA) indicated that the dataset was not suitable for dimensionality reduction, with a Kaiser–Meyer–Olkin (KMO) value of 0.432 and a significant Bartlett’s test of sphericity (χ^2^ = 612, df = 28, *p* < 0.001). A subsequent, more targeted PCA—focused on variables with potential intercorrelation—still showed no meaningful associations between IgE levels and the selected inflammatory markers (C3, C4, fibrinogen, ESR, CRP) despite several parameter adjustments.

The test results indicate that four extracted components meet the criteria of having a correlation coefficient of 1 or higher. The PCA was conducted to investigate the relationships between IgA and IgE levels assessed during two distinct phases (Phase 1 and Phase 2). The Kaiser–Meyer–Olkin (KMO) measure of sampling adequacy was 0.491, suggesting a relatively low level of adequacy for conducting factor analysis. Nonetheless, the results of Bartlett’s Test of Sphericity were statistically significant (χ^2^ = 1314.192, df = 6, *p* < 0.001), indicating the presence of correlations among the variables and validating the suitability of employing principal component analysis (PCA). Based on the Kaiser criterion (eigenvalues greater than 1) and the scree plot, two principal components were retained, which together account for a substantial portion of the total variance. The first component showed strong loadings for IgE in both phases (IgE_Ph1 = 0.766; IgE_Ph2 = 0.767), suggesting that it represents a predominantly IgE-related pattern. The second component was characterized by moderate to high loadings for IgA in both phases (IgA_Ph1 = 0.737; IgA_Ph2 = 0.707), indicating a pattern driven primarily by IgA variation, though still influenced by IgE. The initial component exhibited robust associations with IgE variables across both phases (IgEph1 = 0.766; IgEph2 = 0.767), suggesting that it predominantly embodies an IgE-centered pattern. The second component exhibited moderate to high loadings from IgA variables (IgAph1 = 0.737; IgAph2 = 0.707), suggesting a pattern that is more specific to IgA variation while still being affected by IgE. A robust and statistically significant positive correlation was identified between IgA levels in Phase 1 and Phase 2 (r = 0.972, *p* < 0.001), suggesting that individuals exhibiting elevated IgA levels in Phase 1 were likely to sustain comparable levels in Phase 2. Similarly, IgE levels in Phase 1 and Phase 2 showed an almost perfect positive correlation (r = 0.999, *p* < 0.001), indicating a significant stability of IgE concentrations across the phases. Our analysis revealed that while IgE levels did not correlate meaningfully with inflammatory markers, both IgA and IgE levels demonstrated high internal consistency over time, indicating stability in individual immune profiles. Two distinct immune response patterns emerged—one primarily driven by IgE and the other by IgA—suggesting that these immunoglobulins may act independently in the context of HSP, with limited overlap in their association with systemic inflammation. These findings are visually represented in Figure 3, which illustrates the component loadings and clustering patterns within the dataset.

#### 2.2.2. The Statistical Analyses of IgA and IgE Serum Concentrations (g/L) for Both Patient Cohorts During Both Phases

To investigate potential immunological variations, two peripheral blood samples were collected from each patient: the first at hospital admission (Phase 1) and the second 30 days post-discharge (Phase 2). Patients were subsequently categorized into two groups: those diagnosed with a concurrent infectious disease and those without any infectious diagnosis at the time of admission. Analysis of the immunological panel revealed that 54.34% of the total cohort presented elevated serum IgA levels, while 55.79% demonstrated increased IgE levels. Notably, among patients with a secondary infectious diagnosis, 66.23% exhibited elevated IgE concentrations.

A paired sample *t*-test was conducted to determine whether statistically significant differences existed between the mean serum levels of immunoglobulins (IgA and IgE) recorded during the two collection phases. The results revealed substantial differences in immunoglobulin concentrations between the initial (Phase 1) and follow-up (Phase 2) measurements. Specifically, the *p*-values for all comparisons were less than 0.05, indicating that the observed changes over time were statistically significant. Furthermore, the 95% confidence intervals for the differences did not include zero, reinforcing the reliability of these findings. Although all paired comparisons reached statistical significance, a notable discrepancy was observed between the mean differences of certain groups. Pair 1, representing the change in IgA levels in patients with a secondary infectious diagnosis, showed a relatively small mean difference of 0.04593 (*p* = 0.041).

In contrast, Pair 2, which assessed IgA changes in patients without such a diagnosis, demonstrated a substantially larger mean difference of 0.23983 (*p* = 0.009). Similarly, Pair 3 and Pair 4 evaluated IgE levels in patients with and without secondary infectious diseases, respectively. The mean difference in IgE for Pair 3 was −1.59322 (*p* = 0.022), whereas for Pair 4, the mean difference was markedly greater at −10.99458 (*p* < 0.001). These results are illustrated in Figure 4 and summarized in Table A3.

Independent samples *t*-tests were performed to compare serum immunoglobulin levels (IgA and IgE) between the two patient groups during the two phases. No statistically significant difference was detected between the groups for IgA at phase 1, t(136) = 1.413, *p* = 0.160, with a 95% confidence interval (CI) for the mean difference spanning from –0.119 to 0.713. At phase 2, a notable difference was seen, t(136) = 2.635, *p* = 0.009, revealing elevated IgA levels in one group, with a mean difference of 0.493 (95% CI: 0.123 to 0.862). Significant variations in IgE levels were seen at both time intervals. In phase 1, the assumption of homogeneity of variances was breached (F = 10.094, *p* = 0.002), and the modified *t*-test revealed a highly significant disparity, t(66.547) = 4.256, *p* < 0.001, with a mean difference of 166.76 (95% CI: 88.53 to 244.98). At phase 2, the assumption of unequal variance was upheld (F = 10.039, *p* = 0.002), and the group difference remained statistically significant, t(66.846) = 4.463, *p* < 0.001, with a mean difference of 176.02 (95% CI: 97.30 to 254.74). The findings demonstrate that although IgA levels distinguished the groups over time, IgE levels persistently exhibited significant differences, indicating a robust correlation with the existence of a secondary infectious disease. The immunological analysis revealed substantial changes in both IgA and IgE serum levels over time, particularly in relation to the presence or absence of a secondary infectious disease. While IgA levels showed moderate variation between groups, IgE levels consistently exhibited significant differences in both study phases, strongly associated with infectious status. These findings suggest a more dynamic and infection-sensitive role for IgE in the immune response of pediatric patients with HSP. The results are presented in Figure 4 and Table A4.

### 2.3. Patients with Infectious Diseases as Secondary Diagnoses

This study employed the Receiver Operating Characteristic (ROC) curve to evaluate the diagnostic efficacy of serum IgE levels in differentiating between individuals with subsequent infection diagnoses after admission and those without such diagnoses. The ROC curve results revealed that IgA levels reached statistical significance, with an AUC of 0.817 and a standard error of 0.031, indicating a practical discriminative ability for diagnosing normal IgE levels in patients with infections, as outlined in Table 1 and illustrated in Figure 5. The Youden’s index is 0.556, with an ROC curve cutoff value of 88.615 g/L, indicating that the test possesses a satisfactory capacity to differentiate between positive and negative cases. Individuals exhibiting elevated IgE levels alongside an infection as a secondary clinical diagnosis were categorized as true positives (53 individuals). In comparison, those with elevated IgE levels without a clinical infectious disease diagnosis were classified as false negatives (26 individuals). True negatives consisted of patients exhibiting normal IgE levels without any infections as secondary clinical diagnoses (53 patients), while false positives comprised patients with normal IgE levels who had infections as secondary diagnoses (8 patients). The calculated sensitivity was 67.1%, while the specificity was 86.9%. The results suggest that the model serves effectively as a screening tool, successfully detecting around 67.1% of positive cases while also demonstrating high reliability as a confirmatory test, accurately identifying about 86.9% of true negative cases. The positive predictive value (PPV) was 86.9%, indicating that over two-thirds of the patients who tested positive had normal IgA levels despite having an infectious disease. The negative predictive value (NPV) was 67.1%, suggesting that over two-thirds of patients exhibiting high IgA levels in their screening tests were not affected by infectious diseases. The ROC curve analysis demonstrated that serum IgE levels possess a good discriminative ability for identifying patients with secondary infectious diagnoses. With an AUC of 0.817, high specificity (86.9%), and moderate sensitivity (67.1%), IgE proved to be a reliable biomarker, particularly for ruling out infection. These findings support the potential clinical utility of IgE as a screening and confirmatory tool in this patient population. The results are presented in Figure 5 and Table 1.

### 2.4. Allergen Panel Results and Treatment Approaches

The study evaluated potential digestive and respiratory allergens in all 138 enrolled patients using the Phadiatop test, a multi-allergen allergosorbent screening tool designed for reliable results. Among the 61 patients with negative test results, the majority showed no sensitization to any allergens in the panel. However, two initially negative patients later tested positive for respiratory allergens, including pollen, epithelium, and animal hair. Additionally, two patients from the negative group reported allergic conditions such as non-specific atopic dermatitis and allergic urticaria. The remaining patients with negative results showed no sensitization to digestive or respiratory allergens. Conversely, patients with positive test results primarily exhibited sensitization to respiratory allergens such as animal hair, pollen, and epithelial cells.

Regarding digestive allergens, elevated IgE levels were associated with sensitivity to crab, shellfish, hazelnuts, cow’s milk, and peanuts. Notably, one patient showed positive reactions to penicillin and xylin, while another tested positive for Isoprinosine despite not being included in the tested panel. A considerable number of patients with positive food allergy tests also had a history of atopic dermatitis and elevated IgE levels. Figure 6 presents the allergen panel results for patients with positive tests.

Regarding treatment, nearly all patients (99.3%) were administered antihistamines, highlighting their crucial importance in managing symptoms. Approximately one-third of the patients (34.26%) received prescriptions for NSAIDs, presumably to manage pain and inflammation. In the cohort exhibiting moderate disease severity, corticosteroids were frequently administered, with 79.02% of individuals undergoing this therapeutic intervention. More than half of the patients (54.54%) received antibiotics, particularly in instances where bacterial infections were either confirmed or suspected to be present. All individuals diagnosed with an infectious disease received antibiotic treatment, selected according to the antibiogram results. The antibiotics most commonly used were azithromycin (33.7%), ceftriaxone (32.6%), and cefuroxime (27.7%). The average duration of antibiotic courses was 7 days, with variations spanning from 3 to 21 days based on clinical requirements.

## 3. Discussion

This study aimed to investigate the immunological profile of pediatric patients diagnosed with Henoch–Schönlein purpura (HSP), with a particular focus on serum IgA and IgE levels and their relationship to disease severity. Conducted as a retrospective hospital-based investigation at a tertiary pediatric center, the study included 138 children stratified into severity groups (mild, moderate, and severe) according to the EULAR/PRINTO/PRES criteria for IgA vasculitis. Patients were further categorized based on serum IgE levels, classified as positive or negative, using age-adjusted reference values. By correlating immunological markers with clinical presentation, our findings offer insights into potential inflammatory and allergic mechanisms underlying disease heterogeneity in HSP.

### 3.1. Demographic Characteristics Correlated with HSP Manifestations

The prevalent etiopathogenic model for IgAV suggests that an abnormal immune response is triggered by various antigenic external stimuli in individuals with genetic susceptibility [9,30]. Our findings provide a more detailed understanding of the impact of IgA vasculitis (IgAV) on children, as well as the factors that may affect their onset and severity. This condition appears to be triggered by an atypical immune reaction, likely initiated by infections or allergens, particularly in children with a genetic predisposition [31,32,33]. The pathogenesis of HSP is closely linked to aberrant glycosylation of IgA1 molecules, particularly a reduction in galactosylation of the hinge-region O-glycans. This results in the formation of galactose-deficient IgA1 (Gd-IgA1), which is recognized as an autoantigen by anti-Gd-IgA1 IgG or IgA antibodies, forming circulating immune complexes [19,34]. These immune complexes activate mesangial cells, complement pathways (especially the alternative and lectin pathways), and recruit neutrophils that mediate vascular damage [35].

Our findings indicate that more than 70% of the children diagnosed were under 10 years old, with the majority of cases observed in the 4 to 7 age range [2]. This developmental stage aligns with the period when numerous children begin their schooling or kindergarten experience, settings where they are inherently more susceptible to various infections [36,37]. A clear seasonal trend was also observed, with most diagnoses occurring in autumn, which aligns with the start of the school year. This temporal association may suggest a possible link between increased exposure to pathogens or environmental allergens and the onset of symptoms [30,38]. In contrast, the lowest number of cases was recorded during the summer, when all admissions were related to relapses rather than new diagnoses. This reduction may hypothetically be attributed to school closures, reduced social contact, or more favorable environmental conditions during the summer months [30,39]. The children’s residence also appeared to be important. A greater number of cases originated from urban areas, exhibiting a prevalence approximately 50% higher than that observed in rural regions. Urban children experienced higher rates of impact and were also more prone to moderate manifestations of the disease. This may be associated with increased population density, heightened exposure to pollutants or infections, or potentially improved access to healthcare services in urban areas [8,14]. Gender differences were also noted, with boys more frequently diagnosed than girls and at a slightly younger average age. Furthermore, boys showed a higher likelihood of developing severe disease manifestations [36]. These patterns may reflect underlying differences in immune responses between the sexes, although further studies are necessary to fully elucidate these mechanisms [2].

### 3.2. Immune Response Profiles in Children with Henoch–Schönlein Purpura and Associated Infections

As previously stated, plasma cells in mucosa-associated lymphoid tissue (MALT) generate IgA, which is found in the nasopharynx, tonsils, and gastrointestinal mucosa [30]. Immune complexes, which include IgA antibodies, are generated in response to antigenic exposure resulting from an infection or medication [1,40]. The investigation analyzed the immune response in the context of infectious diseases, contrasting individuals who were positive for different infections with those who were negative. The immunological response exhibited notable differences across the two phases, highlighting the influence of the time interval and infectious disorders on immunoglobulin levels [2,39,41]. This study investigated the potential relationship between IgA and IgE levels, two essential proteins in the immune system, and prevalent inflammation markers in children diagnosed with HSP. This concentrated examination yielded more distinct insights. The PCA results revealed two primary patterns: The first was predominantly influenced by IgE levels at both time points, indicating that these levels exhibited a remarkably consistent trend throughout the phases [28]. The second pattern exhibited a stronger correlation with IgA levels, although some overlap with IgE remained [42]. Our findings indicate that serum IgA levels are elevated in over half of the patients with HSP, especially among those exhibiting moderate to severe manifestations [11,43].

Additionally, these elevated levels are often observed in cohorts that do not have an infectious disease as a secondary diagnosis. The most notable observation was the remarkable stability of both IgA and IgE levels throughout the study. Children exhibiting elevated levels of immunoglobulins at the beginning of the study tended to maintain similar levels later on, showing strong consistency over time. This stability suggests that IgA and IgE may reflect more enduring immune characteristics rather than short-term changes resulting from inflammation. From a clinical perspective, this could mean that early immunoglobulin levels may help identify children who are more likely to follow a particular immune response pattern—something that could be useful in predicting disease progression or tailoring treatments [16,44,45]. Interestingly, and somewhat unexpectedly, the study did not find any strong links between these immunoglobulins and common markers of inflammation, such as CRP or ESR, even after testing several combinations. The results suggest that while IgA and IgE remain stable over time, they may be influenced by different biological pathways than those driving acute inflammation [2,30,46]. In the case of IgE, for example, the stability could reflect a closer connection to allergic or chronic immune responses rather than immediate infection-related inflammation [16]. These findings provide a new layer of understanding of how immune markers behave and could help refine the monitoring and management of complex cases in pediatric settings.

### 3.3. High IgE Levels Without Allergen Panel Positivity

Elevated serum IgE levels have been reported in subsets of children with HSP, particularly in those with a preceding upper respiratory tract infection or a history of atopy [14,47]. IgE may contribute to disease pathogenesis via mast cell activation, vascular permeability, and eosinophilic inflammation, although this role appears secondary to the central role of IgA. Some authors have proposed that IgE-mediated hypersensitivity may act as an adjuvant stimulus, amplifying immune responses and potentially increasing the risk of relapses [48,49].

Importantly, not all patients with elevated IgE exhibit atopy, suggesting that IgE elevation may also reflect a non-specific polyclonal immune activation in response to infections, particularly those caused by streptococcal antigens [19,42,48,50]. The emergence of the mutation responsible for the excessive production of IgE in the body may also be affected by environmental factors [29,49]. Children in urban areas exhibit variations in their exposure to allergens and their sensitivity levels, influenced by geographical factors [20,51]. This is reflected in our data, where a disproportionate number of IgE-positive cases (82 out of 138) originated from urban areas. Such findings suggest that geographic and environmental factors may interact with genetic predispositions, contributing to immune dysregulation [49,52,53]. To enhance the clinical panel, it is essential to keep total serum IgE levels within normal ranges from an early age. This approach has the potential to improve overall well-being and reduce the risk of severe allergic reactions [49,52]. This has important implications for clinical practice. An elevated IgE serum level (>100 IU/mL) typically suggests the presence of allergies, asthma, eczema, or chronic skin infections. However, most patients in this study with elevated IgE serum levels were found to have HSP associated with infections [28,40,42,51]. This raises important questions about the role of IgE beyond allergic pathways. The decoupling of total IgE from specific allergen sensitivity, as observed here, suggests that total IgE may serve as a general indicator of immune system activation rather than a precise marker of allergy alone. Our findings underscore the importance of monitoring IgE levels early in life—not simply as an allergy marker but as part of a broader immunological profile that could flag patients at risk for complex conditions like HSP. The findings of this study indicate that the connection between asthma, characterized by symptoms and bronchial responsiveness, and total IgE levels operates independently of specific IgE levels for prevalent respiratory allergens [54]. This study also investigated the potential of serum IgE levels as a valuable marker for detecting infections that arise following hospital admission in pediatric patients with HSP. Notably, our ROC analysis showed that IgE had moderate diagnostic value for identifying post-admission infections in pediatric HSP patients. While not definitive on its own, elevated IgE levels were more effective in ruling out infection than confirming it. Clinically, this could help stratify patients by infection risk and prioritize further testing or early intervention. Notably, some children with normal IgE levels still developed infections, highlighting the limitations of relying on a single biomarker [22,42]. The test proved to be more effective in confirming the absence of infection. Instead of identifying every true case, its overall performance suggests that it can serve as a valuable resource in clinical environments [40]. Notably, increased IgE levels are frequently correlated with the occurrence of infection, supporting the notion that variations in this immune marker may indicate an underlying infectious process. Conversely, certain patients exhibiting normal IgE levels subsequently developed infections, underscoring the necessity of not depending exclusively on IgE levels for diagnostic purposes [17,22,33]. IgE should be viewed as an integral component of a comprehensive diagnostic approach that encompasses clinical signs, patient history, and additional laboratory markers [39,54]. Although it may not provide conclusive evidence by itself, it can significantly enhance early detection and aid in directing timely and appropriate management, especially in complex situations where the clinical scenario is ambiguous [37]. Taken together, these findings suggest that IgE should be interpreted within a comprehensive clinical context. Its elevation may reflect underlying immune activity linked to infection, even in individuals without atopy. As such, IgE can be a valuable adjunct in complex diagnostic scenarios, particularly when conventional markers yield inconclusive results, thereby supporting timely and targeted management in pediatric settings where immune dysregulation plays a critical role.

It is noteworthy that within the cohort of patients who obtained negative results on allergen panels, a significant proportion exhibited consistently negative outcomes across both respiratory and digestive tests [43]. Nonetheless, a limited group revealed surprising results—two participants who initially tested negative exhibited sensitivity to respiratory allergens, including pollen, animal epithelium, and hair. This suggests that, in certain instances, sensitization may not be readily observable or that variations in exposure or immune response could affect the test results [29,55]. Furthermore, two patients with negative test results exhibited clinical symptoms of atopic conditions, such as non-specific atopic dermatitis and allergic urticaria, suggesting the potential for non-IgE-mediated mechanisms or limitations in the sensitivity of allergen panels [24,55]. On the other hand, individuals with positive allergen panels exhibited reactivity primarily to respiratory allergens, particularly pollen, animal hair, and epithelial cells. Digestive allergens, specifically crab, shellfish, hazelnuts, cow’s milk, and peanuts, were identified in a subset of cases exhibiting elevated IgE levels. A consistent observation among individuals with verified food allergies was a simultaneous history of atopic dermatitis and consistently high IgE levels, underscoring the recognized link between atopic disorders and food hypersensitivities [29,55].

### 3.4. Therapeutic Approaches in Managing Pediatric HSP

Several human leukocyte antigen (HLA) polymorphisms have been associated with an increased susceptibility to HSP [34,35,56]. These alleles may modulate antigen presentation and immune responses to mucosal pathogens, favoring IgA class-switching and promoting autoantibody generation [30,50]. Non-HLA genetic variations, such as those affecting cytokine genes (e.g., IL-6, TNF-α) and complement regulatory proteins, have also been proposed as contributing factors in specific populations [31,32].

Understanding these immunopathological mechanisms helps identify biomarkers for disease activity (e.g., Gd-IgA1, IgE, and complement levels). It may offer targets for future therapies (e.g., B-cell modulation, complement inhibition, or immune complex clearance) [9,31,45,57].

From a therapeutic standpoint, antihistamines were prescribed in nearly all cases, highlighting their central role in symptom management, likely targeting pruritus and other histamine-mediated effects [30]. In contrast, the use of NSAIDs appeared more selective, likely aimed at addressing musculoskeletal pain and localized inflammation rather than systemic involvement [50]. Corticosteroids were primarily reserved for patients with moderate disease severity, reflecting clinical recognition of their utility in controlling systemic inflammation and mitigating vascular complications often seen in HSP [35,37]. Over 50% of the group underwent antibiotic treatment, usually directed by clinical judgment or verification of a bacterial infection. In cases with confirmed infections, the selection of antibiotics was customized according to antibiogram findings, with azithromycin, ceftriaxone, and cefuroxime emerging as the predominant agents utilized. This tailored approach reflects an important aspect of pediatric infectious disease management, where empiric treatment must often be balanced with pathogen-specific strategies to avoid unnecessary antibiotic exposure.

The results underscore the intricate nature of addressing HSP in children, especially in differentiating between IgE-mediated allergic reactions and various inflammatory stimuli [57,58,59]. Although a significant number of individuals in the cohort exhibited increased IgE levels, not every patient showed sensitization to allergens as indicated by standardized panels. This discrepancy highlights the importance of a comprehensive clinical assessment that extends beyond laboratory testing, particularly in the management of children presenting with overlapping allergic and infectious symptoms [2,39]. This disconnect suggests that IgE elevations may reflect broader immune activation rather than classic allergic responses. Clinically, this complicates the differentiation between allergic and infectious triggers, especially in cases where symptoms overlap. As such, our data highlight the importance of a nuanced, individualized clinical approach that integrates laboratory findings with a comprehensive assessment of the patient’s presentation. Overreliance on IgE levels or isolated test results may lead to misinterpretation, whereas a holistic evaluation can provide a more accurate guide to treatment decisions in this heterogeneous population.

One significant limitation of this study is its retrospective design, which restricted the measurement of IgA and IgE levels to only two time points: once during hospital admission, following the onset of clinical symptoms, and again post-treatment during follow-up. This limited temporal resolution constrains our ability to draw robust conclusions about longitudinal trends or dynamic fluctuations in immunoglobulin levels. For instance, elevated or normal IgA concentrations could have been transiently influenced by intercurrent or recurrent infections, which were not systematically captured. A more comprehensive approach, including pre-, intra-, and post-infection sampling, would be required to clarify causal relationships and immune trajectories. This investigation took place in a pediatric clinic and involved a thorough assessment of clinical observation sheets. The study did not include an analysis of IgA subcategories, socioeconomic status, or ethnic origin of the patients, as this information had not been recorded before the study. Socioeconomic status can impact several aspects of pediatric health, including exposure to infections and environmental risks. Its absence limits our ability to assess potential confounding factors or health disparities within the cohort. The retrospective nature of the study also introduces the possibility of information bias, such as missing or incomplete medical records, which may affect the accuracy of the clinical data and laboratory parameters analyzed.

In addition, the study was limited by the lack of IgA subclass analysis, which could have provided greater insight into specific immune responses. These limitations are mainly due to the retrospective nature of data collection, which relied on existing clinical records that did not routinely capture these variables.

## 4. Materials and Methods

### 4.1. Study Design

The clinical study was conducted as a hospital-based, longitudinal, retrospective study involving 138 children aged 1 to 18 years, all of whom were admitted to the “Grigore Alexandrescu” Emergency Clinical Hospital for Children in Bucharest. Due to its status as both an emergency and university hospital, referrals came from various regions across the nation. However, since the study was conducted at a single institution, the data reflect only the pediatric population within the Romanian healthcare system and may not be generalizable to children with HSP in other countries or healthcare contexts. An extensive search was conducted within the hospital’s electronic medical database for pertinent records covering the period from 1 January 2018 to 31 December 2024. All children were previously referred for assessment due to HSP. The children diagnosed with HSP were categorized based on disease severity utilizing the classification criteria established by the European League Against Rheumatism (EULAR), the Pediatric Rheumatology International Trials Organization (PRINTO), and the Pediatric Rheumatology European Society (PRES) for HSP [6].

The samples were collected from venous blood to quantitatively measure the immunological panels, including IgA and IgE serum levels, as well as C3, C4, C-reactive protein, fibrinogen, ESR, and allergen panels for both respiratory and digestive allergens. A vacutainer, either anticoagulant-free or with separating gel, was used as the collection container. The measurement of immunological panel elements was conducted using an enzyme-linked immunosorbent assay (ELISA) [60,61,62,63]. The allergen panels were assessed using the Phadiatop test, which utilizes a multi-allergen allergosorbent to ensure accurate outcomes [64,65]. Table A2 presents comprehensive information regarding the allergen panels associated with both digestive and respiratory allergens.

The collected patient files were entered into a database. The hospital database was used to collect information, including age, gender, environment, immunological panel (IgA and IgE), C3, C4, C-reactive protein, fibrinogen, and ESR, as well as clinical diagnosis and other associated diagnoses. The data were statistically evaluated, and the statistical tests applied are detailed in the “Descriptive Analysis of the Patients” Series’ section (Section 4.2).

After the diagnosis of HSP, the assessment of disease severity (mild, moderate, or severe) was determined following the guidelines set forth by the EULARPRINTO/PRES. These organizations have established the current diagnostic criteria for systemic IgA vasculitis (IgAV) [4,6,65]. In a pediatric patient presenting with purpura, characterized as round or oval and retiform, predominantly on the lower limbs, the diagnosis is confirmed if at least one of the following four criteria is met: (1) abdominal pain, (2) histologically confirmed IgA deposits, (3) arthritis or arthralgia, or (4) renal impairment [4,6,65].

### 4.2. Descriptive Analysis of the Patients’ Series

#### 4.2.1. Inclusion Criteria

The study included patients of both genders, aged between 1 and 18 years, who were admitted to the Pediatric Clinic of “Grigore Alexandrescu” Emergency Clinical Hospital for Children in Bucharest. The children had a prior diagnosis of HSP and underwent testing for their immunological panel, including IgA and IgE, as well as C3, C4, C-reactive protein, fibrinogen, and ESR. Every patient involved in the study had an infectious disease within the six months preceding their hospitalization. To maintain the study’s integrity, only patients with no prior history of COVID-19 infection were included, given the complex treatment protocols associated with the disease.

We classified “positive patients” as individuals exhibiting IgE serum levels exceeding the upper limit of the age-stratified reference interval for IgE. We characterized “negative patients” as individuals exhibiting normal IgE serum levels consistent with age-specific IgE reference values. The thresholds for IgE serum levels are established based on the ELISA Kit protocol and testing guidelines, as detailed in Table A1.

We characterized “mild” symptoms in children with HSP as the presence of rash/purpura (small, reddish-purple spots commonly found on the lower legs and gluteal area), joint pain (slight discomfort and swelling, typically affecting the knees and ankles), and digestive symptoms (intermittent mild abdominal pain that does not significantly impact appetite or daily activities).

We characterized “moderate” symptoms in children with HSP as those presenting with rash/purpura (more extensive purpura that may extend to the arms, face, and trunk, resulting in moderate discomfort), joint pain (more significant pain and swelling in multiple joints, potentially restricting mobility and daily activities), digestive symptoms (recurring abdominal pain, nausea, and occasional vomiting, which can affect appetite and daily routines), and kidney involvement (the presence of blood or protein in the urine, identifiable through urine tests, indicating moderate kidney involvement).

We classified “severe” symptoms in children with HSP as those characterized by rash/purpura (extensive and painful purpura covering large body areas), joint pain (intense pain and swelling in multiple joints, significantly affecting mobility and daily activities), digestive symptoms (acute abdominal pain, ongoing vomiting, and bloody stools, which may result in dehydration and necessitate medical attention), and kidney involvement (notable kidney damage, resulting in ongoing blood or protein in the urine, with the potential to advance to chronic kidney disease).

#### 4.2.2. Exclusion Criteria

The following patients were excluded from the study: those under 1 year of age, patients without a prior diagnosis of HSP, and individuals who had not undergone an immunological or allergen panel test.

#### 4.2.3. Statistical Analysis of Data

The data were statistically analyzed using IBM’s Statistical Package for the Social Sciences (SPSS) version 29 (2022) and Microsoft Excel 2016 (Redmond, WA, USA). The investigation included descriptive statistics, tests to assess normal distribution (Kolmogorov–Smirnov and Shapiro–Wilks), tests to compare quantitative indicators in different groups (comparison of means), correlation analyses (Pearson correlation coefficient, PCC), ROC curves, positive predictive value (PPV), negative predictive value (NPV), and sensitivity and specificity. The chosen significance level was α = 0.05 and 0.01 for PCC. Thus, if the significance level is not reached for values of *p* < α, the null hypothesis is rejected.

## 5. Conclusions

Our comprehensive five-year review offers valuable insights into the manifestation of HSP in children, highlighting how factors such as age, gender, season, and living environment can influence the disease’s progression. Identifying these patterns enables healthcare professionals to remain vigilant for the signs of HSP, promoting timely interventions and facilitating more personalized care that considers each child’s unique background and risk factors. Our findings highlight the unique behaviors of IgA and IgE in HSP. Although inflammation is pivotal in the disease process, these immune markers appear to exhibit a distinct pattern of progression. The robust stability of IgE, specifically, may warrant increased attention in future investigations, particularly in elucidating individual variations in immune response and disease progression.

## Figures and Tables

**Figure 1 ijms-26-06053-f001:**
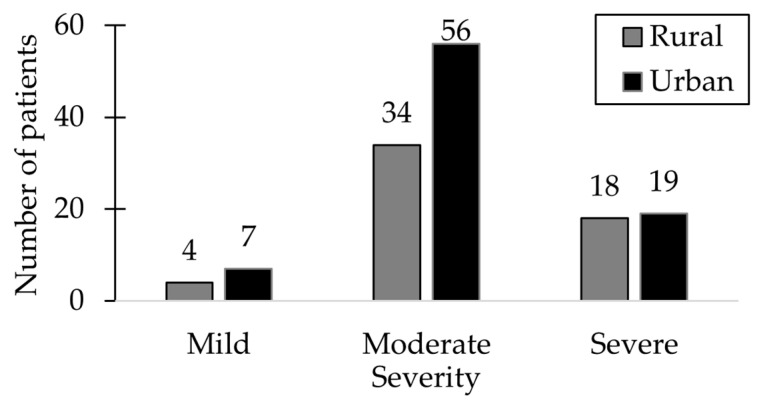
The distribution of patients based on the severity level of HSP and their respective environments.

**Figure 2 ijms-26-06053-f002:**
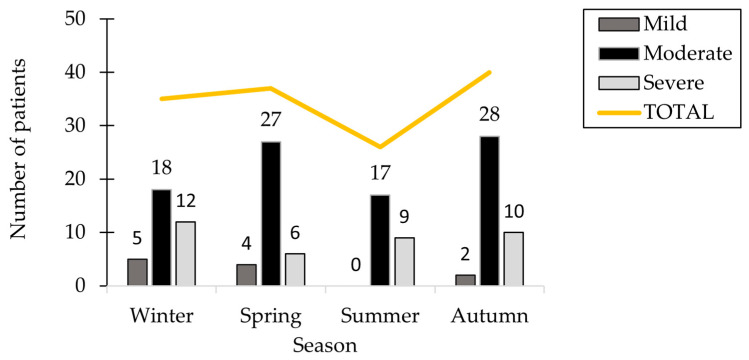
The distribution of patients based on seasonal variations and the severity of HSP.

**Figure 3 ijms-26-06053-f003:**
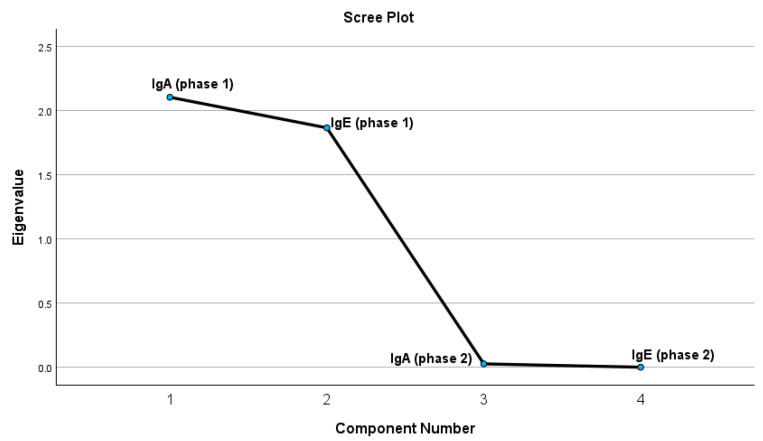
The graphic representation of the PCA test results for possible correlations.

**Figure 4 ijms-26-06053-f004:**
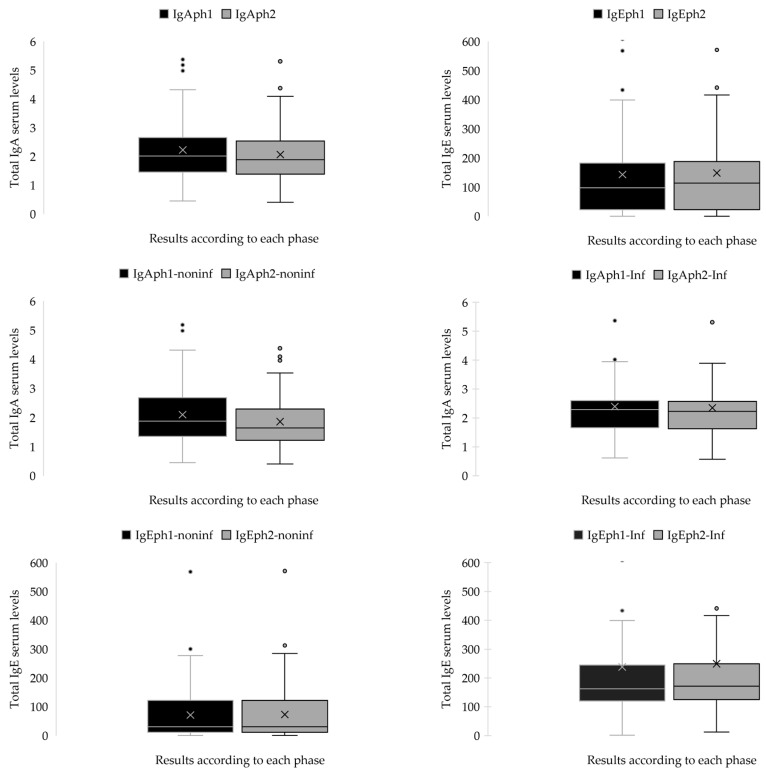
The graphical representation of IgA and IgE serum levels (g/L) for both patient groups across both phases. The circles represent the outliers, and the “x” inside each box denotes the mean value.

**Figure 5 ijms-26-06053-f005:**
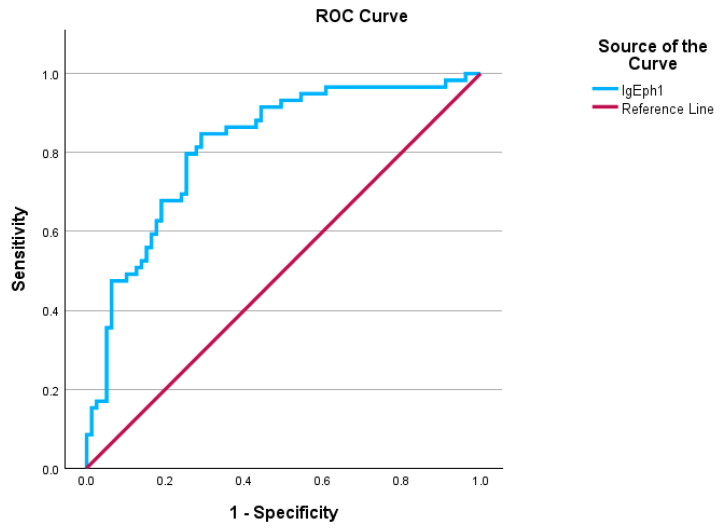
ROC curves obtained from the IgE serum levels. Sensitivity is shown in the ordinate, while the false positive rate (1-specificity) is presented in the abscissa.

**Figure 6 ijms-26-06053-f006:**
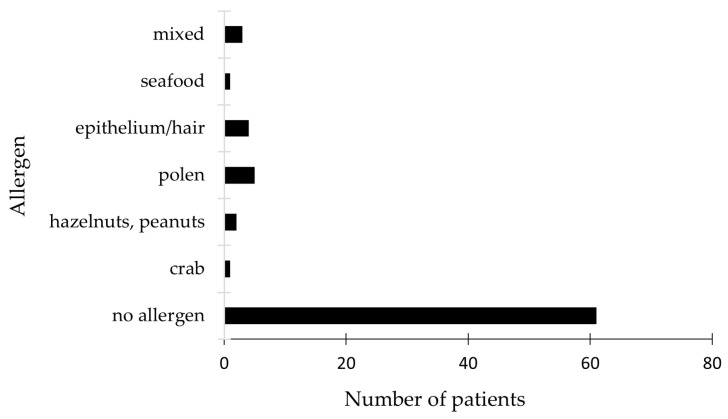
Graphic representation of the allergen panel results for patients identified as having elevated total serum IgE levels.

**Table 1 ijms-26-06053-t001:** ROC analyses for total IgE levels (area under the ROC curve and test result variable(s)).

Area	Std. Error ^a^	Asymptotic Sig. ^b^	Asymptotic 95% Confidence Interval	
			Lower Bound	Upper Bound
0.817	0.031	0.000	0.778	0.907

^a^ Under the nonparametric assumption. ^b^ Null hypothesis: true area = 0.5.

## Data Availability

The data presented in this study are available on request from the corresponding author. The data are not publicly available due to restrictions such as privacy and ethics.

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
