# Peer review of "Serum IgE and IgA Levels in Pediatric Henoch–Schönlein Purpura: Clinical Characteristics and Immunological Correlations in the Context of Infectious Diseases—A Five-Year Retrospective Analysis"

_ijms, 2025, doi:10.3390/ijms26136053_

Round 1

Reviewer 1 Report

Comments and Suggestions for Authors

Based on the 5-year retrospective data of 138 children with HSP, this paper analyzed the relationship between the serum IgA and IgE levels of HSP and infectious diseases by using statistical methods. Meanwhile, the potential influences of demographic factors such as gender, age, and urban and rural environment on patients were investigated. This study is of great value for understanding HSP in children and optimizing clinical management. Suggested for publication.

Several questions:

1)As a retrospective study, please clearly point out the possible information deviations in the data, such as incomplete medical records leading to missing or inaccurate data; In addition, the samples of this study were only from a single hospital and could not fully represent children with HSP in different regions and under different medical conditions. Although they might have been referred, the source of the referral was not clear.

2) Although this article mentions the association between the pathogenesis of HSP and immune response, genetic susceptibility and other aspects, the detailed mechanisms of IgA and IgE in the formation of immune complexes and vascular inflammation, as well as how genetic factors affect children's susceptibility to HSP, are not discussed in depth enough. The theoretical value of this study can be further explored.

Reviewer 2 Report

Comments and Suggestions for Authors

The study entitled “Serum IgE and IgA Levels in Pediatric Henoch–Schönlein Purpura: Clinical Characteristics and Immunological Correlations in the Context of Infectious Diseases – A Five-Year Retrospetive Analysis by Sînziana OpriÈ›escu et al., highlights the role of abnormal immune responses triggered by infections or allergens in genetically predisposed individuals. Most cases occurred in children under the age of ten, particularly during autumn, suggesting that school-related exposure may be a contributing factor. A higher prevalence and severity were observed in urban children, potentially due to environmental conditions or disparities in access to healthcare. Additionally, boys were more frequently affected and exhibited more severe symptoms.

Overall, the manuscript is well written and thoughtfully discussed. While the methodology section describes Phases 1 and 2 of the study and outlines the criteria used to determine HSP severity, these elements appear somewhat disconnected from the Results section. Including a brief summary or reminder of these key methodological points within the Results section would improve clarity and help readers better interpret the findings.

In addition, the authors should report the ethnic origin of the patients included in the study. This information is essential to better understand the potential genetic component of the findings and to evaluate whether similar patterns might be observed in other ethnic groups. Including this detail would enhance the generalizability and interpretability of the results.

The statistical methods used in the study should be accompanied by a simplified interpretation in the Results section. This would greatly benefit readers who are not experts in statistics, helping them to better understand the significance and implications of the findings.

Reviewer 3 Report

Comments and Suggestions for Authors

This retrospective study is based on five years of pediatric clinical records and aimed to evaluate the relationship between the clinical, demographic (age and gender), geographical, and environmental characteristics of children diagnosed with HSP, and the occurrence and progression of the disease. Special attention was given to the role of infectious diseases and IgA and IgE marker levels in the recurrence of HSP.

This study is highly relevant as it provides valuable insights into the potential immunological and environmental triggers of HSP in the pediatric population. The findings suggest that elevated IgE levels, in the context of normal IgA concentrations, are strongly associated with concurrent or recurrent infections in children with HSP, highlighting a possible immunopathological mechanism behind disease exacerbation. These results are highly informative, especially for future diagnostic strategies and for managing and preventing HSP recurrence in children.

However, the manuscript can be revised to benefit from quality improvement.

Abstract:

  • "This study examines the clinical features of children diagnosed with HSP and investigates the association between infectious diseases, elevated IgE serum levels, and the development of HSP."  It seems to me that this part of the aim is not correct; this should be "[...] association between infectious diseases, IgE and IgA serum levels, and the development of HSP."

Introduction

  • L50-57. this paragraph of the introduction is not relevant to the objective of the study. Each word used should be related to the main study, leading to the purpose of the study. Why starting an introduction with IgE? what about IgA? what the impication of IgE in the context of the study? The link between the IgE and HSP is not shown... Please, revise.
  • L59-87 ["The Role Consequences of Elevated Immunoglobulin E (IgE) Levels in the Pediatric Population"]. Same comment. Why talking about IgE? It looks out of context and a flaw from the literature. Its implication with the study purpose is lacking.
  • L88-111. Same Comment. In this paragraph, we don't see IgE no more, but IgA. There is no flow and transition of ideas between the previous and this paragraph. 
  • L112-118. Here, IgE reappears again, without a link with the previous paragraph... Well, needs to rewrite the entire "introduction section."

Results

  • L121-124. "A retrospective hospital-based study was conducted over a five-year period, during which 138 patients were diagnosed with HSP. Out of the total cohort of 138 children, 81 were identified as males, constituting 58.69%, while 57 were identified as females, making up 41.30% of the sample.

This should be rewritten as "A retrospective study conducted from a five-years hospital record included 138 pediatric patients diagnosed with HSP, where 81 (58.69%) were male and 57 (41.30%) were female. Please use this example to rewrite the entire manuscript. Sentences lack grammatical structure; they are unnecessarily long and confusing.

  • L124-126: Same comment; rewrite as "The mean age of the cohort was 7.39 ± 3.86 years old, with the female being slightly older than the male (6.96 ± 0.411 yo vs 8.00 ± 0.536 yo)".

Discussion

  • The discussion section only slightly supports the findings and provides less relevant explanations of the results. “This may be associated with increased population density, heightened exposure to pollutants…” While plausible, these should be framed more clearly as hypotheses unless directly supported by your data.
  • The discussion sometimes sounds like a summary of results rather than an interpretive analysis. For example: “Children exhibiting elevated levels in the initial phase generally sustained comparable levels in the subsequent phase…”
    Instead, you could deepen the interpretation: What does this stability mean for clinical prognosis or pathophysiology?

  • Several concepts are repeated across sections (e.g., the role of IgE in infection detection appears in both 3.2 and 3.3).

  •  "High IgE Levels Without Allergen Panel Positivity" appears 2 times for different paragraphs.
  • Sentences like this appear nearly verbatim twice: “The findings suggest that IgA serum levels generally stay within the normal range for patients admitted with an infectious disease…”

  • The limitations section is appropriately included but should be expanded:
    - For Example, the absence of socioeconomic status data is acknowledged but not discussed in terms of how it might influence outcomes.
    - IgA/IgE measured only twice is a serious limitation, particularly when drawing conclusions about trends.

  • Some long sentences make it harder to grasp the point.

Methods:

  • Can the authors confirm the ethical approval code as "19 from 28 May 2024".

Overall:

  • The grammar is consistently poor, with frequent typographical errors, awkward phrasing, and incorrect verb tenses.

  • Sentences are overly long, convoluted, and difficult to follow, severely impairing readability.

  • The narrative lacks flow and logical progression, with abrupt transitions and redundant information scattered across the discussion.

  • The introduction fails to adequately set the stage for the study.

  • There is a lack of a clear hypothesis or research question.

  • Key background information is either missing or poorly structured, limiting the reader’s understanding of the study’s rationale and objectives.

  • Literature review is flawed and superficial, and important previous work is not properly cited or discussed.

  • No consistent format for reporting statistical results, with some lacking p-values, confidence intervals, or clear comparisons.
  • The role of IgE and IgA is not discussed in a scientifically rigorous manner, leading to potentially misleading conclusions.

  • The potential confounding role of infection on immunoglobulin levels is acknowledged but not adequately controlled for or explored in analysis.

  • Statements about HIES and allergy markers are made without clear diagnostic criteria or robust methodological support.

  • Although the topic is interesting, many of the conclusions are well-established in existing literature and are not presented in a novel or particularly insightful way. In my opinion, the findings do not significantly advance current understanding due to limited depth and analytical rigor.

Comments on the Quality of English Language

See comments to the authors

Round 2

Reviewer 2 Report

Comments and Suggestions for Authors

The authors have made the changes suggested by this reviewer. I have no objection to the acceptance of the revised version for publication.

Author Response

Comment 1: The authors have made the changes suggested by this reviewer. I have no objection to the acceptance of the revised version for publication.

Response 1: We sincerely thank you for your time, constructive feedback, and support throughout the review process. We are grateful that the revised version meets your expectations and we appreciate your recommendation for acceptance.

Reviewer 3 Report

Comments and Suggestions for Authors

This is the second round of review of this manuscript. 

I would like to first thank the authors for their effort to put together and address the comments raised trying to improve their manuscript. 

However, many issues are still to be delicately addressed.

Introduction:

As mentioned in the previous review round, the introduction still looks unnecessarily long. I would urge the authors to shorten it by focusing on the most important data that relate to the scientific question addressed by the paper. I give some (but not all) detailed comments hereafter.

* L53-55. "Systemic IgAV can manifest at any age; however, it is most commonly observed in children between the ages of 3 and 15 years. The data indicate that 90% of disorders beginning in childhood manifest before the age of ten."

Here we have two sentences describing more or less the same thing. These statistics are interesting, but not essential to your immune marker focus. These sentences can be removed or shortened.

* L58. "Pathology is observed in 3–27 cases per 100,000 children, though it is relatively rare in infants."

Epidemiological incidence data can be trimmed unless necessary for context. Also, looks like this sentence does not align with the previous ones (the disease is more prevalent in children, Yes or No?) Make it clear.

* Also, why segmenting the introduction? Why not removing the headlines and make a flowwise introduction to the main purpose?

Results.

* [The mean age of the cohort was 7.39 ± 3.86 years, with females being slightly older than males (6.96 ± 0.411 years vs 8.00 ± 0.536 years). The average age of the group was 7.39 ± 3.86 years.]

The age is in "years old" not "years". Please revise overall.

* Figure 4. Caption: "The graphical representation of IgA and IgE serum levels (g/L) for both patient groups across both phases. The circles represent the outliers." What about the "x" within the boxes?

The manuscript is close to be clean for further consideration. Please, adequately address the above comments. 

Comments on the Quality of English Language

See comments to the authors
